# Patterns of Healthcare Utilization Leading to Diagnosis of Young-Onset Colorectal Cancer (yCRC): Population-Based Case-Control Study

**DOI:** 10.3390/cancers14174263

**Published:** 2022-08-31

**Authors:** Ameer Farooq, Carl J. Brown, Eric C. Sayre, Manoj J. Raval, Jonathan M. Loree, Ria Garg, Mary A. De Vera

**Affiliations:** 1Department of Surgery, Faculty of Medicine, University of British Columbia, Vancouver, BC V6Z 1Y6, Canada; 2Division of General Surgery, St. Paul’s Hospital, Vancouver, BC V6Z 1Y6, Canada; 3Centre for Health Evaluation and Outcome Sciences, St. Paul’s Hospital, Vancouver, BC V6Z 1Y6, Canada; 4Arthritis Research Canada, Richmond, BC V6X 2C7, Canada; 5Department of Medicine, Division of Medical Oncology, Faculty of Medicine, University of British Columbia, Vancouver, BC V5Z 4E6, Canada; 6BC Cancer, Vancouver, BC V5Z 4E6, Canada; 7Faculty of Pharmaceutical Sciences, University of British Columbia, Vancouver, BC V5Z 4E6, Canada; 8Collaboration for Outcomes Research and Evaluation, University of British Columbia, Vancouver, BC V5Z 4E6, Canada

**Keywords:** colorectal cancer, epidemiology, healthcare, case control study

## Abstract

**Simple Summary:**

There is a rising incidence of colorectal cancer among young patients. We attempted to characterize how young colorectal cancer patients use the healthcare system prior to diagnosis to see if there is a potential window where we could diagnose patients earlier. We found that young colorectal cancer patients did not seem to present more frequently than healthy controls in the years leading up to their diagnosis, contrary to prior studies. Other interventions are needed to diagnose yCRC patients earlier.

**Abstract:**

Background: The increasing risk of young-onset colorectal cancer (yCRC) in adults < 50 years has called for better understanding of patients’ pathways to diagnosis. This study evaluated patterns of healthcare utilization before diagnosis of yCRC. Methods: Using linked administrative health databases in British Columbia, Canada, we identified yCRC cases and cancer-free controls matched (1:10) on age, sex, and healthcare utilization. The index date was the date of diagnosis for yCRC cases and matched date for controls. Outpatient visits, emergency department visits, and hospitalizations over a 5-year prediagnosis period (e.g., year-1 to year-5) were compared using descriptive statistics and Poisson regression models. Results: The study included 2567 yCRC cases (49.6% females, 43.0 ± 5.8 years) and 25,455 controls (48.6% females, 43.0 ± 5.8 years). We observed an increasing number of outpatient visits from prediagnosis year-5 (median = 3) to year-1 (median = 8) for yCRC cases. Among controls, outpatient visits were stable and did not have a pattern of increase. Poisson regression models indicated higher adjusted count ratios for outpatient visits for yCRC cases compared to controls in the year before diagnosis (1.11; 95% CI, 1.07 to 1.15). In the year before diagnosis, 35.1% of yCRC cases had potentially related visits to CRC (e.g., nausea, vomiting) and 16.9% had potentially red flag visits (e.g., gastrointestinal hemorrhage or iron deficiency anemia). Conclusions: Using population-based data, we found that individuals with yCRC did not have higher healthcare utilization than individuals without in the prediagnosis period except for the year before diagnosis.

## 1. Introduction

Colorectal cancer (CRC) is the third most diagnosed malignancy in both the United States and Canada [1,2]. In Canada, it is the second and third leading cause of cancer-related death among men and women, respectively [1]. Recent evidence has demonstrated an increase in the incidence of CRC amongst young adults [3,4]. In Canada, the incidence of young-onset CRC (yCRC) diagnosed in adults less than 50 years of age has increased by a mean annual percentage change (APC) of 4.45% for men since 2010 and by 3.47% for women since 2006 [5]. Approximately 10% of cases of CRC in Canada are now diagnosed in individuals <50 years of age [4,5].

This increasing risk of yCRC has called for research to understand care and outcomes of patients. Of particular interest, little is known about patterns of healthcare utilization among patients prior to their diagnosis, particularly at the population-level. Previous studies, mainly in the United States, have suggested that yCRC patients face diagnostic delays [6,7,8,9,10]. Scott and colleagues, for example, found that young patients with rectal cancer had a delay of 217 days compared to 29.5 days for average age rectal cancer patients [8]. However, to our knowledge no studies have specifically investigated if these delays have been found in a single payer healthcare system (such as in Canada). To address these gaps, we conducted a population-based epidemiologic study with the specific objectives of: (1) assessing patterns of healthcare utilization before diagnosis of individuals with yCRC as compared with age- and sex-matched individuals without cancer; (2) assessing presenting complaints before diagnosis of individuals with yCRC as compared with individuals with average-age onset colorectal cancer (aCRC; ≥50 years).

## 2. Methods

### 2.1. Data Source

We used data from a population-based CRC cohort established with administrative databases held in British Columbia (BC), Canada. These include Population Data BC [11], which contains longitudinal and deidentified individual-level health services data for the population of BC (approximately 4.86 million in 2016) [11] since April 1985 including information from the following databases. The Medical Service Plan (MSP) database contains all billings for the province of BC, and would encompass all visits to a physician, whether in-patient or outpatient [12]. The Discharge Abstract Database (DAD) is a nationally created database through the Canadian Institute for Health Information (CIHI) [13]. It captures administrative, clinical and demographic information on hospital discharges (including deaths, sign outs, and transfers). We also were able to obtain vital statistics from the province of BC, which includes all deaths in BC [14]. Additionally, we linked to the National Ambulatory Care Reporting System (NACRS) which contains data for emergency department visits [15]. Finally, the BC Cancer Registry captures all new cancers diagnosed in BC residents since 1985 including information on diagnosis (e.g., date, tumour group, sites) and treatment (e.g., dates, modality) [16].

### 2.2. Study Design

Using these data sources, we conducted a case control study. Cases were defined as individuals with CRC identified in the BC Cancer Registry with the following International Classification of Diseases for Oncology, Third Edition codes (C18.2–C18.9 [colon]; C19.9 [rectosigmoid]; and C20, C21.8 [rectum]) [17] and were diagnosed between 1 January 1999 to 31 December 2016. The rationale for including these individuals is that complete capture of diagnostic information (e.g., using International Classification of Diseases Codes [ICD], 9th Revision) in the databases began on 1 January 1994, thus allowing 5 years of data to assess patterns of presentation before CRC diagnosis. We assigned the index date as the date of definitive diagnosis from the BC Cancer Registry based on tissue diagnosis of CRC (endoscopist, surgeon or oncologist).

Individuals with CRC were matched to cancer-free controls (1 up to 10) based on age, sex and index diagnosis date. The controls were cancer-free throughout the entire study period. Additionally, controls were required to have a healthcare utilization, that is a recorded visit in any of MSP, DAD, or NACRS database within same year of diagnoses of their matched case. Controls were assigned the index date of their matched case. This design was chosen to define a large matched control group with which we could then identify potential associated variables with yCRC healthcare utilization. We used the age of diagnosis variable from the BC Cancer Registry to define yCRC among individuals who received their diagnosis <50 years of age. As well, we defined aCRC as individuals those who received their diagnosis ≥50 years (Figure 1) illustrates the study design and identification of CRC cases and cancer-free controls).

### 2.3. Healthcare Utilization

We assessed healthcare utilization before the index date for yCRC cases and cancer-free controls. Specifically, we created prediagnosis time periods (also interchangeably termed the “prodromal” time period) corresponding to each preceding year from the index date (e.g., year-1 to year-5), as shown in Figure 1. We quantified the number of outpatient visits in MSP, hospitalizations in the DAD, and ED visits in NACRS. We used ICD9 codes in the MSP [18] or and ICD10 in the DAD [19] to characterize reasons for visits as well as identified visits for colonoscopies using fee item codes 33374, 33373, and 10731 in the MSP database. For further context, we additionally assessed patterns of healthcare utilization in the year of diagnosis (i.e., year 1), that is the 1 year period following the index date (this period is also shown in Figure 1).

**Figure 1 cancers-14-04263-f001:**
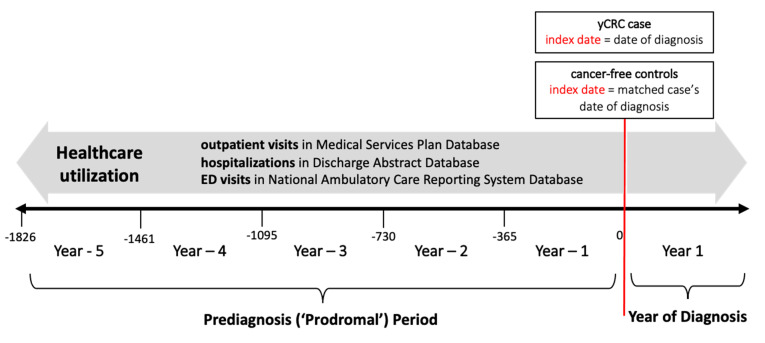
Schematic of study design in timing of assessment of patterns and cohort selection of healthcare utilization.

### 2.4. Presenting Complaints

We characterized presenting complaints for outpatient visits among individuals with yCRC and aCRC using ICD9 codes in the MSP database. These were then categorized as “potentially related” or “unrelated”, depending on whether they may be linked to a diagnosis of CRC according to guidelines from the UK National Institute for Health and Care Excellence (NICE) [20]. “Potentially related visits” were further identified as “potentially red flag visits” if they were one of the following symptoms found in the NICE guidelines: diagnosis of malignancy, abdominal pain, gastrointestinal hemorrhage or iron deficiency anemia [20]. Of note, anemia was a diagnosis made based on ICD codes, not on the actual lab value itself.

### 2.5. Statistical Analysis

To achieve our first objective, we used descriptive statistics (e.g., mean, median) to assess patterns of healthcare utilization in terms of outpatient visits, hospitalizations, and ED visits for each year during the prediagnosis period (e.g., year-1 to year-5) and in the year of diagnosis (e.g., year 1) for individuals with yCRC and cancer-free controls. We additionally characterized outpatient visits and hospitalizations according to months (e.g., prediagnosis month-60 to -12) in order to have greater granularity. Furthermore, for outpatient visits, within each prediagnosis year (e.g., year-5, year-4, year-3, year-2, and year-1, we used dispersion-corrected Poisson regression [21] to evaluate the association between healthcare utilization and yCRC diagnosis. Models were only applied for outpatient visits given the high frequency of zero counts for hospitalizations and ED visits. The additional dispersion correction accommodated deviations from the Poisson assumption of variance to the mean and exponentiated regression coefficients are interpreted as expected count ratios associated with yCRC diagnosis. Models were adjusted for demographic characteristics including age, sex, socio-economic status (using a proxy measure based on neighbourhood income quintile), and residence (rural versus urban, as determined by using Census Metropolitan Area/Census Agglomeration from geographical census data). We also considered as covariates comorbid conditions including inflammatory bowel disease (IBD), anxiety, depression, diabetes, and hyperlipidemia and Romano Charlson comorbidity index, defined in the year before each prediagnosis year modelled. Variables representing comorbid conditions were entered in a stepwise manner.

To achieve our second objective, we used descriptive statistics (e.g., counts and proportions) to assess the frequency of presenting complaints for outpatient visits before diagnosis for individuals with yCRC and aCRC. We used Chi-square tests for comparisons. All analyses were conducted using SAS Version 9.4 (SAS Institute Inc., Cary, NC, USA). This study was approved by the University of British Columbia (H17-03530). All inferences, opinions, and conclusions drawn in this manuscript are those of the authors, and do not reflect the opinions or policies of the Data Steward(s).

## 3. Results

### 3.1. Demographics

The study sample included 2567 (49.6% females; 43.0 ± 5.8 years) yCRC cases with a mean age at index date (e.g., diagnosis) of 43.0 ± 5.8 years and 25,455 matched cancer-free controls (48.6% females) and with a mean age at index date of 43.0 ± 5.8 years. Table 1 shows characteristics of yCRC cases and cancer-free controls as well as aCRC cases and their cancer-free controls. The distribution of tumor site in yCRC patients was predominantly rectal (41.6%) and left sided (40.5%). Individuals with yCRC and cancer-free controls had similar frequency of comorbid conditions except for IBD with a higher proportion among yCRC than cancer-free controls (6.8% vs. 0.9%).

### 3.2. Health Care Utilization

Table 2 summarizes health care utilization visits among yCRC cases and cancer-free controls during prediagnosis period (year-1 to year-5) and year of diagnosis (year 1) in terms of median and mean number of outpatient visits, hospitalizations, ED visits. With respect to outpatient visits for yCRC cases, we observed an increasing number of visits from prediagnosis year-5 (median = 3) to year-1 (median = 8). Among controls, outpatient visits were stable and did not have a pattern of increase. These findings are further reflected in multivariable Poisson regression models on the association between outpatient visits and yCRC diagnosis (Table 3). yCRC cases had lower expected adjusted count ratios for outpatient visits compared to cancer-free controls patients from prediagnosis year-5 (86; 95% confidence interval [CI], 0.82 to 0.90) to year-2 (0.81; 95% CI, 0.77 to 0.84), which changed to a higher expected adjusted count ratio in year-1 (1.11; 95% CI, 1.07 to 1.15). This trend continued into the year of diagnosis (year 1) where an adjusted count ratio of 2.42 (95% CI, 2.35 to 2.49) indicated a 2-fold higher frequency of visits among yCRC cases as compared to cancer-free controls. Finally, when we characterized health care utilization visits according to months, for greater granularity, we found the greatest increase in visits occurred in the 2 months before yCRC diagnosis (Figure 2). In additional assessment of health care utilization among yCRC and aCRC cases during the same period, we found lower number of outpatient visits among yCRC in year-5 to year-1 (Appendix A). However, there was a higher average number of ED visits for yCRC cases compared to aCRC cases in prediagnosis year-1 (0.212 vs. 0.164 visits per year, *p* = 0.0054).

### 3.3. Presenting Complaints

Based on patterns of healthcare utilization, we focused our assessment of presenting complaints in the year before diagnosis (prediagnosis year-1), and comparing individuals with yCRC and aCRC. During prediagnosis year-1 we found a higher proportion of yCRC patients with potentially red flag (16.9% vs. 9.4%, *p* < 0.001) and potentially related (35.5% vs. 29.2%, *p* < 0.001) visits compared to aCRC patients. With respect to specific complaints, we found a higher proportion of yCRC visits for nausea and vomiting (14.9% vs. 10.1%, *p* < 0.001), abdominal pain (6.7% vs. 3.0%, *p* < 0.001), and hemorrhoids (3.2% vs. 1.4%, *p* < 0.001) compared to aCRC visits. In contrast, we found a lower proportion of yCRC visits presenting with “other disorders of the intestine” (5.5% vs. 6.6%, *p* < 0.001) (Table 4 and Appendix A). Notwithstanding our focus in prediagnosis year-1, we also assessed presenting complaints in earlier prediagnosis years, with our findings suggesting lower proportion of patients presenting with potentially related visits to CRC (Appendix A).

## 4. Discussion

Using a population-based administrative health data, we assessed patterns of healthcare utilization and presenting complaints before diagnosis among individuals with yCRC to inform identification of potential diagnostic opportunities. This is relevant given that the increasing incidence of yCRC [4] has prompted interest in optimal strategies for timely diagnosis, particularly as prior studies have reported diagnostic delays among yCRC patients [6,7,8,9,10].

To our knowledge, this is the first study to evaluate the pathway to diagnosis for patients with yCRC at a population level in a single-payer, national healthcare system. With respect to patterns of healthcare utilization over the 5-year prediagnosis period, outpatient visits are the most frequent, with hospitalizations and ED visits occurring to a much lesser extent. When compared to age- and sex-matched cancer-free controls, we did not find yCRC cases to have higher healthcare utilization in prodromal years except for the year before diagnosis (i.e., year-1), where we identified an uptick in patterns of healthcare utilization particularly for outpatient visits, but also for hospitalizations and ED visits. These findings persisted in multivariable models that adjusted for sociodemographic characteristics and comorbidities. These may suggest that there may not be missed diagnostic opportunities for yCRC in earlier prodromal years (e.g., years-5 to year-2), when patients do not seem to be experiencing symptoms that necessitate healthcare utilization. Nonetheless, as visits to family physicians represent the majority of outpatient visits in the MSP database, this highlights where awareness and education on the increasing risk of yCRC may be targeted, with implications for both diagnosis and outcomes of yCRC. Though caution in applying to yCRC must be taken as the sample was limited to individuals aged > 67 years, 2013 case–control study of Medicare beneficiaries demonstrated the association between primary care utilization before diagnosis and lower CRC incidence and improved outcomes, namely lower mortality [22].

To complement our assessment of patterns of healthcare utilization, we characterized presenting complaints for yCRC patients during prediagnosis years, particularly for outpatient visits by assessing ICD-9 codes. Our results suggest that when presented, symptoms are being recognized by healthcare providers. This provides reassurance compared to prior findings, particularly survey data from other jurisdictions, that individuals with yCRC felt that their symptoms were being minimized or dismissed [9]. Nonetheless, practitioners should continue with efforts to raise awareness around the symptoms and risk factors and have a high index of suspicion for CRC diagnosis, even for young patients. Of note, 3.2% of outpatient visits for yCRC patients were for hemorrhoids in prediagnosis year-1 compared to 1.4% of aCRC patients. In a review of 55 papers examining yCRC patients, the two most common symptoms at presentation were rectal bleeding and abdominal pain, which is not dissimilar to aCRC patients [23]. In 2015, the National Institute for Health and Care Excellence in the UK published guidelines recommending that any patient aged under 50 with rectal bleeding and any of the following symptoms should be referred to the suspected cancer pathway: abdominal pain, change in bowel habit, weight loss, iron deficiency anemia [20]. In a publicly funded system such as Canada and the UK, appropriate use of referrals is an important consideration to minimize waste and unnecessary consultation. However, our data would support vigilance in a young patient with symptoms compatible with CRC, especially in the 40- to 50-year age range. In particular, flexible sigmoidoscopy should be considered in young patients even with seemingly benign disease, given the higher rates of left sided and rectal cancers in young patients [24].

Our work has implications for improving the care of yCRC patients. More outreach and education should be done to educate young adults on the signs and symptoms of CRC, with an emphasis on the rising incidence of yCRC. This may improve the chances of young patients presenting earlier to a healthcare provider when they first experience symptoms and minimizing the chances of diagnosis at a more advanced stage. Unfortunately, CRC continues to be an underpublicized disease that is underreported in the media [25]. As we describe earlier, it is important for practitioners to have awareness around the symptoms and risk factors and have a high index of suspicion for CRC diagnosis, even for young patients. This is particularly important given that recommendations for the age of initiating screening for CRC remains a complex issue. The US Preventative Services Task Force recently lowered the age for screening for average risk patients to age 45 [26]. Results on the effectiveness of decreasing the age for eligibility for screening has been mixed [5,27,28]. Abualkhair et al. found a drastic increase of CRC patients as individuals shifted from the 49 to 50 years old, suggesting a large number of preclinical undetected yCRC cases that might be detected with a lower age for guideline-based screening [28]. Our group similarly found a steady increase through this same age transition [5]. Modeling of CRC screening strategies in Canada suggest that stool-based screening would yield 20 additional life-years per 1000 people screened along with a 10% increase in colonoscopy demand, compared to ages 50 to 74 [27]. Costs would also increase by 13% and 14% for fecal immunochemical testing and fecal occult blood testing, respectively [27]. Thus, the cost-effectiveness of lowering the age cut-offs and the ability of the healthcare system to accommodate the increase in colonoscopies is unclear [29]. A more cost-effective approach might be a targeted screening program that considers family history and other known risk factors for yCRC, in addition to the institution of universal tumor testing for microsatellite instability [29]. Future steps should ultimately include a prospective cohort study of yCRC patients similar to the Reducing the bUrden of Breast cancer in Young women (RUBY) study [30] to comprehensively investigate individual personal and tumor risk factors, diagnostic pathways, and outcomes.

Study strengths and limitations warrant discussion. Our study cohort was drawn from source population over 30 years created by linking data from Population Data BC and the BC Cancer Registry, which captures data on approximately 95% of all cancer cases in the province. The BC Cancer Registry is reviewed annually for quality, completeness, and accuracy by the North American Association of Central Cancer Registries [16]. Nonetheless, this is a retrospective study utilizing administrative data, which has inherent limitations with respect to the accuracy of the data and changes in coding over the study period. As well, the BC Cancer Registry lacks sufficient data on CRC disease stage, which was collected beginning in 2012 and is not acquired using a systematic approach as information on stage relies on a variety of data sources such as death certificates, pathology reports, and death certificates. We considered known confounders that might make patients present to a healthcare provider, but certainly did not capture all possible confounders, especially those that cannot be measured using administrative data.

## 5. Conclusions

Using generalizable, population-based data, we found that individuals with yCRC did not have higher healthcare utilization than cancer-free controls in the prediagnosis period except for the immediate year before diagnosis. Further efforts to lower the burden of yCRC should focus on public and patient outreach, physician education, as well as the consideration of more targeted screening interventions.

## Figures and Tables

**Figure 2 cancers-14-04263-f002:**
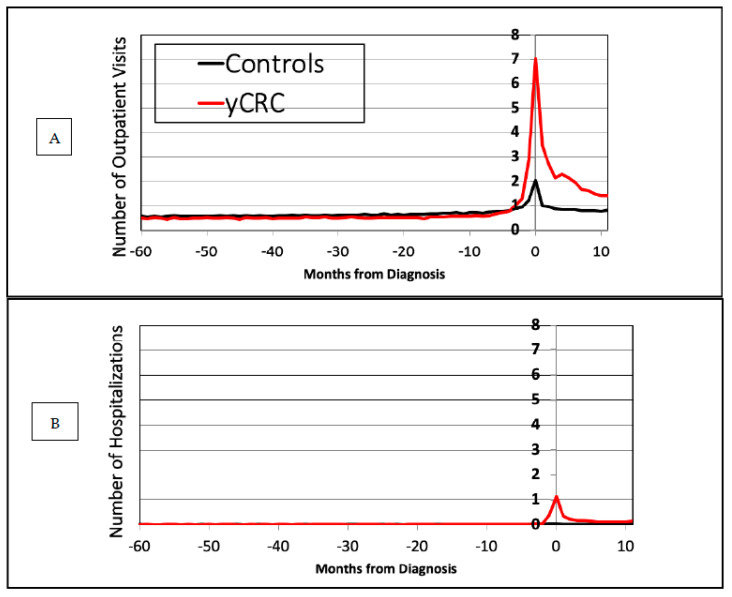
Patterns of healthcare utilization in terms of (**A**) outpatient visits and (**B**) hospitalizations for individuals with young-onset CRC and cancer-free controls (y-axis denotes number of visits and x-axis denotes months, with month 0 indicating ‘index date’).

**Table 1 cancers-14-04263-t001:** Characteristics of individuals with young-onset colorectal cancer (yCRC; <50 years), average-onset colorectal cancer (aCRC; ≥50 years) and respective cancer-free controls.

	yCRC Cases (*n* = 2567)	Controls (*n* = 25,455)	aCRC Cases (*n* = 32,014)	Controls ^A^ (*n* = 297,399)
Age at diagnosis (years, mean ± SD)	43.0 ± 5.8	43.0 ± 5.8	69.2 ± 9.8	69.2 ± 9.9
Age at diagnosis (groupings)	<30: 91 (3.5%) 30–39: 474 (18.5%) 40–49: 2002 (78.0%)	<30: 1133 (4.5%) 30–39: 4706 (18.5%) 40–49: 19,616 (77.1%)	50–59: 6010 (18.8%) 60–69: 10,096 (31.5%) 70–79: 11,144 (34.8%) ≥80: 4764 (14.9%)	50–59: 56,959 (19.2%) 60–69: 92,464 (31.1%) 70–79: 103,663 (34.9%) ≥80: 44,225 (14.9%)
**Sex**				
Female	1273 (49.6%)	12,381 (48.6%)	15,151 (47.3%)	140,549 (47.3%)
Male	1294 (50.4%)	13,074 (51.4%)	16,863 (52.7%)	156,762 (52.7%)
**Neighbourhood Income Quintile**				
Quintile 1	483 (18.8%)	5300 (20.8%)	6981 (21.8%)	62,612 (21.1%)
Quintile 2	492 (19.2%)	5307 (20.9%)	6399 (20.0%)	59,850 (20.1%)
Quintile 3	551 (21.5%)	5055 (19.9%)	6360 (19.9%)	58,072 (19.5%)
Quintile 4	549 (21.4%)	5183 (20.4%)	6127 (19.1%)	56,952 (19.2%)
Quintile 5	492 (19.2%)	4610 (18.1%)	6147 (19.2%)	59,825 (20.1%)
**Residence**				
Rural	319 (12.4%)	2760 (10.8%)	4708 (14.7%)	41,373 (13.9%)
Urban	2248 (87.6%)	22,695 (89.2%)	27,306 (85.3%)	255,938 (86.1%)
**Comorbidities ^B^**				
Hypertension	235 (9.2%)	2663 (10.5%)	12,447 (38.9%)	109,285 (36.8%)
Anxiety	51 (2.0%)	656 (2.6%)	518 (1.6%)	3863 (1.3%)
Depression	249 (9.7%)	3421 (13.4%)	2419 (7.6%)	20,577 (6.9%)
Diabetes	155 (6.0%)	1462 (5.7%)	6310 (19.7%)	48,965 (16.5%)
Dyslipidemia	73 (2.8%)	836 (3.3%)	5391 (16.8%)	49,169 (16.5%)
Inflammatory Bowel Disease	174 (6.8%)	217 (0.9%)	690 (2.2%)	1555 (0.5%)
**Tumor Site**				
Left Colon	1040 (40.5%)		13,192 (41.2%)	
Right Colon	260 (10.1%)		5088 (15.9%)	
Transverse Colon	122 (4.8%)		1854 (5.8%)	
Rectal	1067 (41.6%)		10,347 (32.3%)	
Unknown	78 (3.0%)		1533 (4.8%)	

^A^ Cancer-free controls for individuals with aCRC were not analyzed for study purposes but reporting demographic characteristics for completeness; ^B^ Comorbidities defined in the year before index date (for purposes of reporting demographic characteristics.

**Table 2 cancers-14-04263-t002:** Healthcare utilization among yCRC cases and cancer-free controls during prediagnosis period (years-5 to -1) and year of diagnosis (year 1).

	Prediagnosis Period					Year of Diagnosis
	Year-5	Year-4	Year-3	Year-2	Year-1	Year 1
**Outpatient visits**						
yCRC (median)	3	3	4	4	8	25
yCRC (mean)	5.80 ± 8.06	5.85 ± 8.29	6.08 ± 9.01	6.23 ± 9.26	10.80 ± 10.16	29.41 ± 21.21
Control (median)	4	4	4	5	6	7
Control (mean)	6.72 ± 9.70	6.99 ± 10.21	7.28 ± 10.65	7.76 ± 11.44	9.64 ± 12.99	11.0 ± 14.87
**Hospitalizations**						
yCRC (median)	0	0	0	0	0	3
yCRC (mean)	0.13 ± 0.44	0.14 ± 0.49	0.15 ± 0.52	0.15 ± 0.54	0.60 ± 0.88	2.95 ± 1.89
Control (median)	0	0	0	0	0	0
Control (mean)	0.13 ± 0.49	0.14 ± 0.49	0.14 ± 0.49	0.14 ± 0.51	0.20 ± 0.67	0.25 ± 0.70
**Emergency visits**						
yCRC (median)	0	0	0	0	0	0
yCRC (mean)	0.0051 ± 0.098	0.022 ± 0.44	0.035 ± 0.44	0.051 ± 0.37	0.21 ± 0.73	0.48 ± 1.42
Control (median)	0	0	0	0	0	0
Control (mean)	0.0044 ± 0.081	0.024 ± 0.37	0.043 ± 0.48	0.067 ± 0.56	0.12 ± 0.75	0.15 ± 0.90

**Table 3 cancers-14-04263-t003:** Multivariable Poisson regression models showing association between diagnosis of yCRC and outpatient visits during prediagnosis period (years-5 to -1) and year of diagnosis (year 1).

	Count Ratio (95% Confidence Interval)					
	Prediagnosis Period					Year of Diagnosis
	Year-5	Year-4	Year-3	Year-2	Year-1	Year 1
yCRC (ref: no yCRC)	0.86 (0.82 to 0.90)	0.84 (0.80 to 0.88)	0.83 (0.80 to 0.87)	0.81 (0.77 to 0.84)	1.11 (1.07 to 1.15)	2.42 (2.35 to 2.49)
Age ^A^	0.99 (0.98 to 0.99)	0.98 (0.98 to 0.99)	0.98 (0.98 to 0.99)	0.98 (0.98 to 0.98)	0.98 (0.98 to 0.98)	0.99 (0.99 to 0.99)
Female (ref: male)	1.35 (1.31 to 1.38)	1.32 (1.29 to 1.35)	1.27 (1.24 to 1.30)	1.20 (1.18 to 1.24)	1.11 (1.08 to 1.13)	1.04 (1.02 to 1.13)
Neighbourhood income						
Quintile 1 (ref: Quintile 5)	1.28 (1.23 to 1.34)	1.25 (1.20 to 1.30)	1.30 (1.25 to 1.35)	1.36 (1.30 to 1.41)	1.31 (1.27 to 1.36)	1.31 (1.27 to 1.35)
Quintile 2 (ref: Quintile 5)	1.13 (1.08 to 1.18)	1.14 (1.09 to 1.19)	1.11 (1.07 to 1.16)	1.20 (1.15 to 1.25)	1.16 (1.11 to 1.20)	1.13 (1.09 to 1.17)
Quintile 3 (ref: Quintile 5)	1.13 (1.08 to 1.17)	1.10 (1.06 to 1.15)	1.06 (1.02 to 1.11)	1.11 (1.07 to 1.16)	1.11 (1.07 to 1.15)	1.10 (1.06 to 1.14)
Quintile 4 (ref: Quintile 5)	1.07 (1.02 to 1.12)	1.02 (0.98 to 1.07)	1.03 (0.99 to 1.08)	1.06 (1.01 to 1.10)	1.03 (0.99 to 1.07)	1.02 (0.98 to 1.05)
Rural residence (ref: urban)	0.84 (0.80 to 0.87)	0.85 (0.82 to 0.89)	0.87 (0.83 to 0.90)	0.86 (0.82 to 0.89)	0.88 (0.85 to 0.91)	0.89 (0.86 to 0.92)
Comorbidities						
Hypertension (ref: no)	1.48 (1.40 to 1.56)	1.51 (1.44 to 1.58)	1.47 (1.41 to 1.54)	1.39 (1.33 to 1.45)	1.32 (1.27 to 1.37)	1.24 (1.20 to 1.28)
Anxiety (ref: no)	1.48 (1.37 to 1.60)	1.57 (1.47 to 1.68)	1.64 (1.52 to 1.75)	1.55 (1.45 to 1.66)	1.45 (1.37 to 1.54)	1.42 (1.35 to 1.50)
Depression (ref: no)	2.08 (2.01 to 2.15)	1.98 (1.92 to 2.04)	1.99 (1.93 to 2.05)	2.05 (1.99 to 2.11)	1.83 (1.78 to 1.88)	1.72 (1.67 to 1.76)
Diabetes (ref: no)	1.60 (1.50 to 1.71)	1.55 (1.46 to 1.65)	1.55 (1.47 to 1.64)	1.45 (1.37 to 1.53)	1.45 (1.39 to 1.52)	1.39 (1.33 to 1.44)
Dyslipidemia (ref: no)	1.32 (1.20 to 1.45)	1.46 (1.35 to 1.58)	1.42 (1.32 to 1.53)	1.39 (1.30 to 1.49)	1.16 (1.09 to 1.24)	1.12 (1.06 to 1.18)
IBD (ref: no)	1.83 (1.64 to 2.03)	1.86 (1.68 to 2.06)	1.98 (1.79 to 2.19)	1.84 (1.66 to 2.04)	1.74 (1.59 to 1.90)	1.32 (1.24 to 1.41)
Charlson comorbidity score ^A,B^	1.24 (1.21 to 1.28)	1.29 (1.26 to 1.32)	1.32 (1.29 to 1.35)	1.30 (1.28 to 1.33)	1.28 (1.26 to 1.30)	1.06 (1.05 to 1.07)

^A^ Modelled as continuous variable. ^B^ Comorbidities defined in the preceding year for each prediagnosis year modelled.

**Table 4 cancers-14-04263-t004:** Most frequent presenting complaints * for outpatient visits during prediagnosis year-1 for yCRC and aCRC cases.

	yCRC		aCRC		*p*-Value
	*n*	%	*n*	%	
Unrelated visits (1)	6536	48.1	147,682	61.4	<0.001
Potentially related visits (2)	4769	35.1	70,119	29.2	<0.001
Potentially red flag visits (3)	2293	16.9	22,587	9.4	<0.001
**Specific complaints**					
Nausea & vomiting	2029	14.9	24,177	10.1	<0.001
Abdominal pain	914	6.7	7138	3.0	<0.001
“Other disorders of intestine” OR “Other symptoms including abdomen and pelvis”	742	5.5	15,946	6.6	<0.001
Hemorrhoid	438	3.2	3397	1.4	<0.001
Anemia	304	2.2	4977	2.1	0.19
Rectal bleeding	241	1.8			<0.001
**Total visits**	**13,598**		**240,388**		

* Recorded ICD9 codes in the MSP database were used to identify reasons for visits (‘complaints’). All percentages are out of entire subset of presenting complaints for the respective age cohort. These were then mapped as: (1) unrelated visits for symptoms that may not be connected to a diagnosis of CRC; (2) potentially related visits for symptoms that may be connected to a diagnosis of CRC (nausea & vomiting, other abdominal or pelvic symptoms); (3) potentially red flag visits which include a diagnosis of a malignancy, abdominal pain, gastrointestinal hemorrhage, or iron deficiency anemia, consistent with NICE guidelines. Please refer to Appendix A for ICD9 codes and descriptions.

## Data Availability

The data that support the findings of this study are available from Population Data BC (https://www.popdata.bc.ca/, accessed on 2 June 2021) but restrictions apply to the availability of these data, which were used under license for the current study, and so are not publicly available. Data are available from Population Data BC through a data access request (dataaccess@popdata.bc.ca).

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
