# Peer review of "Patterns of Healthcare Utilization Leading to Diagnosis of Young-Onset Colorectal Cancer (yCRC): Population-Based Case-Control Study"

_cancers, 2022, doi:10.3390/cancers14174263_

Round 1

Reviewer 1 Report

General comments. Thank you for the opportunity to review this manuscript. The authors address a timely issue to investigate pre-diagnosis health resource use for patients with yCRC, using a case control study design. In general, this manuscript would benefit from a clearer description of the evidence gap that this work intends to address, as well as greater methodological transparency and justification of decisions. Specific comments are provided below:

Introduction

-       The introduction would benefit from a clearer illustration of the unmet need, and why the authors have chosen to look at pre-diagnosis healthcare utilization patterns. Why is this appropriate and what can the findings be used to inform? The evidence gap is currently unclear. 

-       The implications of the work are made clearer in the discussion section, but the intro would be strengthened if this were described earlier.

Methods

-       Please justify the use of 1:10 matching and describe the matching method that was used

-       How were matching variables selected and why were variables limited to age and sex? 

-       Did the authors consider other matching variables?

-       Please clearly describe which types of data are included in each of the accessed databases (e.g. DAD/NACRS)

-       It would be helpful to more clearly define the control cohort (e.g. did they remain free of CRC or any cancer throughout the entire study period?)

-       Please explain how the final regression model was built (e.g. how were relevant covariates determined, was collinearity considered and addresses, interaction effects?)

Discussion

-       Suggest avoiding the term “universal healthcare system” when speaking about Canada, as this is not entirely accurate. 

Author Response

General comments. Thank you for the opportunity to review this manuscript. The authors address a timely issue to investigate pre-diagnosis health resource use for patients with yCRC, using a case control study design. In general, this manuscript would benefit from a clearer description of the evidence gap that this work intends to address, as well as greater methodological transparency and justification of decisions. Specific comments are provided below:

Introduction

-    The introduction would benefit from a clearer illustration of the unmet need, and why the authors have chosen to look at pre-diagnosis healthcare utilization patterns. Why is this appropriate and what can the findings be used to inform? The evidence gap is currently unclear. 

Thank you for that feedback. The introduction has been made clearer to reflect the need to look at pre-diagnosis healthcare utilization patterns (page 2, paragraph 2):

“Previous studies, mainly in the United States, have suggested that yCRC patients face diagnostic delays (17 - 21). Scott and colleagues, for example, found that young patients with rectal cancer had a delay of 217 days compared to 29.5 days for average age rectal cancer patients (19). However, to our knowledge no studies have specifically investigated if these delays have been found in a single payer healthcare system (such as in Canada). “

The implications of the work are made clearer in the discussion section, but the intro would be strengthened if this were described earlier.

Thank you for this suggestion, and we have made the necessary changes as noted above on page 2, paragraph 2.

Methods

Please justify the use of 1:10 matching and describe the matching method that was used

Thank you for the question.  The matched controls were identified as part of a larger CIHR funded grant, entitled the Examining epidemiology, treatment, and outcomes in young-onset colorectal cancer (EXPLAIN-yCRC). Patients were matched based on age, sex, and index date. We did not match on other variables such as comorbidity as those were explanatory variables we felt were of interest in our analysis. This allowed us to generate a large matched group of young patients without cancer to help us find potential associated characteristics of yCRC patients. To date, very little is known about the risk factors for yCRC and so attempting to match on other variables would have been fraught with confounding. The manuscript was edited on page 2, paragraph 5 to better explain this.

How were matching variables selected and why were variables limited to age and sex? 

Thank you for the excellent question.  As mentioned above, we did not match on other variables such as comorbidity as those were explanatory variables we felt were of interest in our analysis. This allowed us to generate a large matched group of young patients without cancer to help us find potential associated characteristics of yCRC patients. To date, very little is known about the risk factors for yCRC and so attempting to match on other variables would have been fraught with confounding.

Did the authors consider other matching variables?

Other matching variables had been considered when the EXPLAIN database was first created, but as there is currently little known about risk factors for yCRC, other matching variables were not used. Other determinants of health such as geographic area and income were used in subsequent analysis but not in the matching process.

Please clearly describe which types of data are included in each of the accessed databases (e.g. DAD/NACRS)

Thank you for the clarification. We have more clearly outlined what data is included in each of the accessed databases on page 2, paragraph 3.

It would be helpful to more clearly define the control cohort (e.g. did they remain free of CRC or any cancer throughout the entire study period?)

The control cohort remained free of any cancer diagnoses throughout the entire study period. We have made that clearer on page 2, paragraph 5.

Please explain how the final regression model was built (e.g. how were relevant covariates determined, was collinearity considered and addresses, interaction effects?)

In computing our multivariable Poisson regression model, we adjusted for demographic characteristics (age, sex, socioeconomic status using a proxy measure based on neighbourhood income quintile, and residence) as these are known determinants of healthcare utilization. Of note, given our use of administrative health data, we were limited by available variable but nonetheless, aforementioned variables that we were able to include are key determinants of health. Also as described in page 4, we also considered covariates comorbid conditions as these would also impact encounters with the healthcare system.  These were entered in a stepwise manner and only variables with significant associations with our study outcome were included in the final model to achieve parsimony.  We did not assess collinearity and interaction effects.  We provided further detail on our model building approach in page 4. 

Discussion

Suggest avoiding the term “universal healthcare system” when speaking about Canada, as this is not entirely accurate. 

Thank you for this comment. This term was removed throughout the manuscript (page 8, paragraph 2; page 9, paragraph 2).

Reviewer 2 Report

Congratulations for the hard work. The subject seems to be of potential future interest. However, important issues and limitations exist:

- introduction is too short and not so relevant to the subect

- there are no conclusions

- moderate english language revisions are necessary 

- the limitations mentioned in the text are important. I would suggest reducing the research period and eliminate some of these drawbacks (e.g. coding, cancer staging)

Author Response

Congratulations for the hard work. The subject seems to be of potential future interest. However, important issues and limitations exist:

- introduction is too short and not so relevant to the subject

Thank you for that comment. The introduction was revised to provide more rationale and an understanding of the impetus for the study (page 2, paragraph 2).

- there are no conclusions

Thank you for noting this. A conclusion section was added (page 10, paragraph 2).

- moderate english language revisions are necessary 

Thank you for the comment – an effort was made to go through the document and revise any excessive wordiness or grammatical errors. We would be open to make any specific changes that are requested.

- the limitations mentioned in the text are important. I would suggest reducing the research period and eliminate some of these drawbacks (e.g. coding, cancer staging)

Thank you for this comment. Inherent to any database study are the limitations with respect to what data is available as well as the vagaries of coding. Unfortunately we do not have access to staging data for our cancer cohort currently.

Round 2

Reviewer 2 Report

Congratulations for the modifications.  In the current form your manuscript can be considered for acceptance.